# Surgical Resection of a Recurrent Hepatocellular Carcinoma with Portal Vein Thrombosis: Is It a Good Treatment Option? A Case Report and Systematic Review of the Literature

**DOI:** 10.3390/jcm11185287

**Published:** 2022-09-07

**Authors:** Giuseppe Sena, Daniele Paglione, Gaetano Gallo, Marta Goglia, Mariasara Osso, Bruno Nardo

**Affiliations:** 1Department of Vascular Surgery, Pugliese-Ciaccio Hospital, 88100 Catanzaro, Italy; 2Department of Pharmacy, Health and Nutritional Sciences, University of Calabria, 87036 Rende, Italy; 3Department of Surgical Sciences, Sapienza University of Rome, 00161 Rome, Italy; 4Department of General Surgery, Sant’Andrea University Hospital, Sapienza University of Rome, 00189 Rome, Italy

**Keywords:** hepatocellular carcinoma, portal vein tumor thrombosis, surgical resection

## Abstract

Background: Hepatocellular carcinoma (HCC) is the sixth most frequent diagnosed tumor worldwide and the third leading cause of cancer related death. According to the EASL Guidelines, HCC with portal vein tumor thrombosis (PVTT) is classified as an advanced stage (BCLC stage C) and the only curative option is represented by systemic therapy. Therefore, treatment of HCC patients with PVTT remains controversial and debated. In this paper, we describe the case of a 66-year-old man with a recurrent HCC with PVTT who underwent surgical resection. A systematic review of the literature, comparing surgical resection with other choices of treatment in HCC patients with PVTT, is reported. Methods: A systematic review of the literature regarding all prospective and retrospective studies comparing the survival outcomes of HCC patients with PVTT treated with surgical resections (SRs) or other non-surgical treatments (n-SRs) has been conducted. Case presentation: A 66-year-old Caucasian man with a history of Hepatitis C Virus (HCV) related liver cirrhosis and previous hepatocellular carcinoma of the VI segment treated with percutaneous ethanol infusion (PEI) seven years before presented to our clinics. A new nodular hypoechoic lesion in the VI hepatic segment was demonstrated on follow-up ultrasound examination. A hepatospecific magnetic resonance imaging (MRI) scan confirmed also the presence of a 18 × 13 mm nodular lesion in the V hepatic segment with satellite micronodules associated with V–VIII sectoral portal branch thrombosis. The case was then discussed at the multidisciplinary team meeting, and it was decided to perform a right hepatectomy. The postoperative course was regular and uneventful, and the discharge occurred seven days after the surgery. At eight-month follow-up, there was no clinical nor radiological evidence of neoplastic recurrence, with well-preserved liver function (Child-Pugh A5). Results: Nine studies were included in the review. Median Overall Survaival (OS) ranged from 8.2 to 30 months for SRs patients and from 7 to 13.3 for n-SRs patients. In SR patients, one-year survival ranged from 22.7% to 100%, two-year survival from 9.8% to 100%, and three-year survival from 0% to 71%. In n-SRs patients, one-year survival ranged from 11.8% to 77.6%, two-year survival from 0% to 47.8%, and three-year survival from 0% to 20.9%. Conclusion: The present systematic literature review and the case presented demonstrated the efficacy of surgery as a first-line treatment in well-selected HCC patients with PVTT limited or more distal to the right and left portal branches. However, further studies, particularly randomized trials, need to be conducted in future to better define the surgical indications.

## 1. Introduction

Hepatocellular carcinoma (HCC) is the sixth most frequent diagnosed tumor worldwide and the third leading cause of cancer related death. Moreover, it has shown a progressive increase in its incidence and mortality rate in the last few years [1,2]. HCC prognosis has improved due to advances in diagnosis and treatment, however overall survival rates appear substantially unchanged over the past twenty years. Since the diagnosis is very often delayed, the incidence of vascular invasion is still elevated. HCC usually tends to infiltrate the portal venous system. The incidence of portal vein tumor thrombosis (PVTT) ranges between 44.3% and 62.4% [3] and is higher than thrombosis of hepatic veins/vena cava and bile ducts (0.7–20% and 1.84–13% respectively) [3,4,5]. PVTT is associated with jaundice, portal hypertension, ascites, and distant metastases, significantly worsening the prognosis of the patient with HCC, which is approximately 2.7 months. There are several choices of treatment available for HCC such as surgical resection, liver transplantation, arterial transcatheter chemoembolization (TACE), radiofrequency ablation (RFA), external radiotherapy (RT), transcatheter arterial radioembolization (TARE), hepatic arterial infusion chemotherapy (HAIC), and therapy with Sorafenib [6]. According to the European HCC Guidelines which have accepted the Barcelona Clinic Liver Cancer Staging (BCLC) system, HCC with PVTT is classified as an advance stage (BCLC stage C) and the only therapeutic option is represented by systemic treatment, which consists of first-choice first-line treatment in the combination of Atezolizumab with Bevacizumab [7]. However, several authors, particularly the oriental ones, very often reported having a more aggressive approach in an attempt to make the classic “stage hierarchy” approach less rigid in favor of a more rational “therapeutic hierarchy” approach [8]. In particular, the recommendations of the Asian Pacific Association for the Study of the Liver (APASL) suggest surgery as a potentially radical treatment of HCC with PVTT [9]. Moreover, the Chinese guidelines on the diagnosis and treatment of primary liver cancer also propose surgery in the HCC patient with PVTT as a therapeutic option [10].

Therefore, treatment of HCC patients with PVTT remains controversial and debated. In this paper, we described the case of a 66-year-old man with recurrent HCC with PVTT treated by surgical resection. We also conducted a systematic review of the literature comparing the efficacy of surgical resection and other treatment modalities in the management of HCC patients with PVTT.

## 2. Materials and Methods

### 2.1. Study Selection

We performed a systematic literature review of all prospective and retrospective studies comparing the survival outcomes of HCC patients with PVTT treated with surgical resections (SRs) or other therapies, including TACE, RFA, HAIC, TARE systemic chemotherapy, best-supportive care and other non-surgical treatments (n-SRs). The literature search was conducted in accordance with the Preferred Reporting Items for Systematic Reviews and Meta-analyses (PRISMA) standards [11]. We searched the following electronic databases for studies up to 19 January 2022 (Pubmed, Scopus, Web of science, Embase, Medline, Cochrane Library and Google scholar), using the following keywords: “Surgical resection”, “Hepatocellular carcinoma”, “Portal vein tumor thrombosis”. The full text papers were evaluated individually by two authors (SG and DP). The Pubmed “related articles” function was used to enhance the search and the references of each potentially eligible article were evaluated. A manual search was conducted to reduce the finding bias. Only studies in English were selected regardless of the ethnicity of the study population. The final eligibility decision was obtained from the consent of the two authors who evaluated the papers. The case report is developed according to the CARE checklist [12].

### 2.2. Inclusion and Exclusion Criteria

In our review, we have included: (1) prospective or retrospective studies evaluating the outcomes of SRs vs. n-SRs in HCC patients with PVTT, (2) studies reporting overall survival (OS) for at least three years, (3) studies which simultaneously compared SRs vs. n-SRs. Studies that met the following criteria were excluded: (1) patients with metastatic tumors, (2) adjuvant or neoadjuvant therapy associated with surgical resections, (3) no survival report, (4) lack of simultaneous comparison between SRs vs. n-SRs.

### 2.3. Data Extraction

Two authors independently extracted the data, and a third author has revised them. Possible disagreements were solved through a collegial meeting. All data were reported on a collection form. The research and data extraction were conducted using the Population, Intervention, Control, Outcome (s) (PICO) search framework. The content of this framework is outlined below:Population: HCC patients with PVTT.Intervention: All curative surgical resection.Control: Patients treated with other therapies including TACE, RFA, HAIC, TARE systemic chemotherapy, best-supportive care and other non-surgical treatments.Outcome: Median Overall Survival, 1-, 2-, 3-year Median Overall Survival.

Data regarding the characteristics of the studies were extracted: year of publication, surname of the first author, country, number of patients in the SRs and n-SRs groups, mean age, gender of patients, median overall survival, 1-, 2-, 3-year Median Overall Survival, study design. The characteristics of the patients considered were: number of patients in each Child-Pugh class, number of patients with or without portal hypertension, number of patients with different levels of alpha fetoprotein (AFP), number of patients with or without hepatitis B virus (HBV) and number of patients with different types of PVTT. Cheng’s classification was used to graduate the extent of PVTT: Type I, tumor thrombi in segmental or sectoral branches; Type II, tumor thrombi in the right or left portal branches; Type III, tumor thrombi in the portal trunk; Type IV, tumor thrombi in the portal trunk [13].

### 2.4. Statistical Analysis

The risk of bias in non-randomized Studies of interventions (ROBINS-I) was used to assess the risk of bias [14]. The (ROBINS-I) assesses the risk of bias for quantitative studies comparing the effectiveness of an intervention on two groups of patients. The tool is based on different domains: confounding bias, selection bias, bias in the classification of interventions, bias due to the deviation from the intended interventions, bias due to lost data, bias in the measurement of results, bias in data report. Each of these domains can be assessed as: ‘low risk’, ‘moderate risk’, ‘serious risk’, ‘critical risk’, or ‘no information’. Statistical analysis was performed only on the data of the selected studies. The descriptive statistics (simple counts, percentages and means) was used to summarize the characteristics of the studies including OS, median overall survival rates at 1, 2, and 3 years. A meta-analysis could not be performed due to the great heterogeneity of the selected studies.

## 3. Results

### 3.1. Case Presentation

A 66-year-old Caucasian man presented a recurrent nodular hypoechoic lesion of the VI hepatic segment on follow-up ultrasound examination. The patient reported to have a positive past medical history for chronic ischemic heart disease and aortic valve insufficiency treated by coronary artery bypass graft (CABG) and aortic valve replacement, diabetes mellitus, arterial hypertension, and symptomatic cholelithiasis treated by laparoscopic cholecystectomy about 15 years ago. About seven years before the US examination, the patient was diagnosed with a HCV related liver cirrhosis and previous segment VI hepatocellular carcinoma treated with Percutaneous ethanol infusion (PEI). Physical examination showed no abnormality. The patient had an excellent performance status: Eastern Cooperative Oncology Group (ECOG) 0. Furthermore, he was classified as A5 according to the Child-Pugh score and the MELD score was 8/9. Routine laboratory investigations detected high blood glucose levels (136 mg/dL), alanine amino transferase (58 U/L), total amylase (104 U/L), pancreatic amylase (76 U/L) and normal platelet count (225 × 10^6^/mL). Whole-body conventional computed tomography (CT) revealed 17 × 12 mm nodular lesion in the V hepatic segment with contrast enhancement compatible with hepatocellular carcinoma and right portal branch thrombosis with slight contrast enhancement as from neoplastic thrombosis. However, distant metastases were not identified (Figure 1). A hepatospecific magnetic resonance imaging (MRI) scan confirmed the presence of a 18 × 13 mm nodular lesion in the V hepatic segment with satellite micronodules associated with V-VIII sectoral portal branch thrombosis (Figure 2). In the hepatospecific phase, the absence of contrast washout from the V-VIII segments, suggested their neoplastic involvement. The Future Remnant Liver Volume (FRLV) (S1-S2-S3-S4), calculated with the “Hepatic VCAR” segmentation software on the Whole-body CT scan, was 50% (Figure 3). Esophagogastroduodenoscopy (EGDS) was unremarkable. The multidisciplinary team, considering the MELD score, the Child class, the FLRV and the patient’s will, indicated a major liver resection. On intraoperative ultrasound, the lesion affecting the V hepatic segment was confirmed. Furthermore, neoplastic thrombosis of V-VIII sectoral portal branch was identified with extension up to 1.5 cm from the origin of the right portal branch. A right hepatectomy was then performed. The portal vein stump was closed with 4.0 prolene running suture. The final histopathological report was consistent with the moderately differentiated hepatocellular carcinoma with trabecular and multinodular features (Edmonson–Steiner grade 2/3). Lymph vascular invasion was present while the perineural one was absent. Margins of the resected specimen were free from neoplastic invasion (T2 Nx). The molecular pattern resulted as follows: Heppar −/+, Glypican +, MOC31−, Cytokeratin 19. Satellite micronodules in V segment presented the same histological aspects. In the V-VIII sectoral portal branch and in the right portal branch localization of hepatocarcinoma with trabecular aspects. The length of the surgery was 325 min. The postoperative course was regular and uneventful, with resumption of nutrition on the second postoperative day and removal of drains on the fourth and sixth postoperative day. The discharge occurred seven days after the surgery. At eight-month follow-up, there was no clinical nor radiological evidence of neoplastic recurrence, with well-preserved liver function (Child A5).

### 3.2. Results of the Systematic Review

The study screening process was conducted in accordance with the Preferred reporting items for systematic reviews and meta-analyzes [11] and is summarized in the PRISMA flowchart shown in the Figure 4. The initial search found 315 records. After the removal of the duplicates and the evaluation of the titles/abstracts, 33 studies were identified for the full text analysis. Of these, 26 were excluded as they did not meet the inclusion criteria (lack of overall survival data n. 11, lack of comparison between SRs and n-SRs n. 6, patients treated with other therapies alongside surgery n. 9). Finally, nine studies were included in the review [15,16,17,18,19,20,21,22,23]. Of the 8261 patients included in the evaluation, 2746 underwent SRs and 5515 received n-SRs. In all studies, patients underwent both major and minor resections, and non-anatomical resections. Furthermore, the resection margin was adequate in all patients. In four papers the n-SRs patients received only TACE, one manuscript reported only systemic therapy and in the remaining different therapies were reported, including TACE, RFA, HAIC, Sorafenib, and best supportive care. The characteristics of the studies are summarized in Table 1. Most of the studies come from Eastern countries. Indeed, six papers are Chinese and two Japanese. Only one comes from the USA. The studies were published between 2005 and 2020. Baselines and characteristics of patients who were enrolled in the included studies are summarized in Table 2. According to the ROBIN-I checklist, one study, five studies, and three studies are judged to have respectively low, moderate and serious risk of bias. The risk of bias are shown in Table 3. None included patients with distant metastases. The degree of PVTT was reported in five studies, persistence of HBV infection was reported in eight studies. Only three papers indicate the presence of portal hypertension. Seven studies reported simultaneously the median survival and the survival rate at one, two, and three years, only one registered the median survival, and only another one the survival rate at one, two, and three years alone. Median OS ranged from 8.2 to 30 months for SRs patients and from 7 to 13.3 for n-SRs patients. In SR patients, one-year survival ranged from 22.7% to 100%, two-year survival from 9.8% to 100%, and three-year survival from 0% to 71%. In n-SRs patients, one-year survival ranged from 11.8% to 77.6%, two-year survival from 0% to 47.8%, and three-year survival from 0% to 20.9%. In addition, three studies reported a statistically significant increase in survival rate only in SRs patients with PVTT type I/II, and a comparable survival rate between SRs patients and n-SRs patients with HCC with PVTT type III/IV [17,18,19].

## 4. Discussion

PVTT is considered one of the worst prognostic factors in HCC patients. These patients are generally treated with an N-SR approach [24,25,26,27]. According to Barcelona Clinic Liver Cancer Staging (BCLC) system, systemic therapy with different drug combinations represents the only therapeutic option [28,29]. However, several authors and in particular Shi et al. suggest that SR, in HCC patients with PVTT, reduces portal pressure, prolongs survival and improves liver function and patients’ quality of life [30]. Several previous studies have demonstrated the safety and efficacy of SRs in well-selected HCC patients with PVTT. A mean survival ranging from 8.9 to 33 months has been reported in surgically treated HCC patients with PVTT [31,32,33,34,35].

A meta-analysis including 160 HCC patients with PVTT demonstrated that surgical treatment was more effective than TACE for type I/II PVTT. However, for patients with type III/IV PVTT there were no significant differences between surgical treatment and TACE [36]. Therefore, surgical treatment is indicated in HCC patients with type I/II PVTT, PST grade 0–2 or Child-Pugh level A or normal liver function (IGC15), resectable primary tumor, absence of intrahepatic or distant metastases. At present the five-year survival rate is 10–59% for type I/II PVTT and 0–26.4% for type III/IV PVTT.

The type of surgical approach depends on the degree of PVTT according to Cheng’s classification.

A segmental hepatectomy is sufficient for a type I, a hepatectomy is needed for a type II a, and extensive hepatectomy may be necessary for a type IIb. Hepatectomy plus thrombectomy or “en bloc” resection with portal vein reconstruction is required for types III/IV. Thrombectomy is performed through an incision of the portal wall after liver resection with or without peeling of the inner side of the portal wall. When the thrombus invades the portal wall or when removal is difficult, resection with portal end to end reconstruction is required. Several studies have compared the different surgical approaches and have shown that there are no significant differences for overall survival and disease-free survival at one, three, and five years between hepatectomy plus thrombectomy and “en-bloc” resection with portal reconstruction [37,38].

Another study demonstrated that five-year survival is not reduced if an “en-bloc” resection is performed [39].

However, the decision on the type of surgical approach depends on several factors, such as the surgeon’s experience with portal reconstructions in the nature of the thrombus. Indeed, there are no randomized trials comparing the two approaches.

A resection margin of less than 1 mm is considered a negative prognostic factor, but its significance remains controversial [37]. Furthermore, there may be various methods to increase the survival in HCC patients with PVTT undergoing surgical resection. In particular, TACE after surgery can reduce the relapse rate and increase survival time [40,41,42] and, in addition, oral administration of Sorafenib, as demonstrated by the Eastern Hepatobiliary Surgical Hospital study, may also increase the survival rate in HCC patients with PVTT after surgery [43]. In this study, we presented the case of a 66-year-old patient with HCC with PVTT involving V–VIII sectoral portal branch treated with a right hepatectomy. According to Cheng’s classification it was a type I PVTT. Because the PVTT involved only the V–VIII sector branch with extension up to 1.5 cm from the origin of the right portal branch without involvement of the right portal branch, portal trunk, or beyond, a right hepatectomy was considered adequate. The postoperative course was uneventful, and the patient is currently disease free. At the same time, we conducted a systematic review of all the studies in the literature that compared surgical resections with all other therapies in HCC patients with PVTT. It was not possible to perform a meta-analysis due to the strong heterogeneity of the selected studies. In most of the studies analyzed, overall survival was significantly higher in SRs patients than in n-SRs sand in particular in HCC patients with type I/II PVTT, consistently with the literature so far produced. However, there are several limitations in this study. First, all SRs were performed in high-volume centers for HBP disease. Second, most of the selected studies are retrospective and not randomized trials. Third, the patients selected in the studies were highly heterogeneous. Fourth, the risk of bias of the selected studies was rated as low to serious. Finally, most of the selected papers were produced in Asian countries. Therefore, the results of this study in relation to the important risk of bias and heterogeneity, must be interpreted with caution.

## 5. Conclusions

The present systematic literature review and the case presented demonstrated the efficacy of surgery as a first-line treatment in well-selected HCC patients with PVTT limited or more distal to the right and left portal branches. However, further studies, particularly randomized trials, need to be conducted in future to better define the surgical indications.

## Figures and Tables

**Figure 1 jcm-11-05287-f001:**
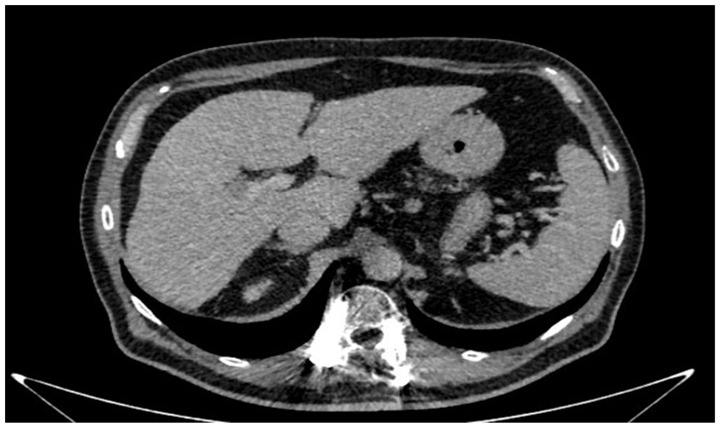
Conventional computed tomography revealed 17 mm × 12 mm nodular lesion in the V hepatic segment.

**Figure 2 jcm-11-05287-f002:**
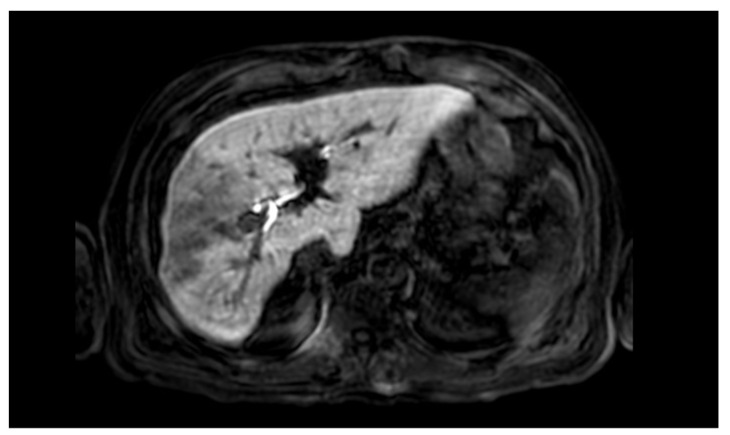
Hepatospecific magnetic resonance imaging scan confirmed the presence of a 18 × 13 mm nodular lesion in the V hepatic segment with satellite micronodules associated with V–VIII sectoral portal branch thrombosis.

**Figure 3 jcm-11-05287-f003:**
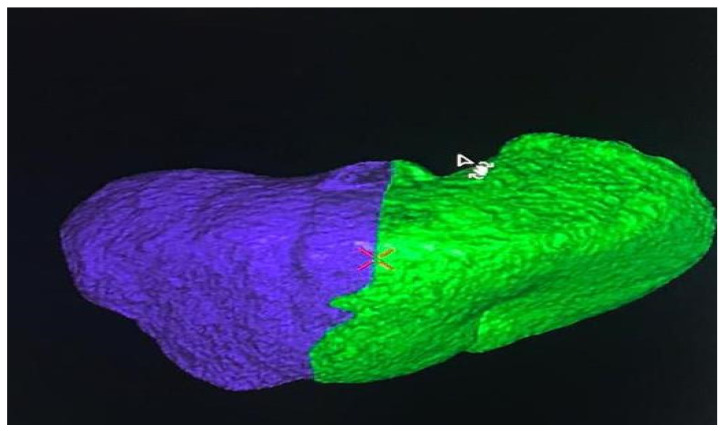
The Future Remnant Liver Volume (FRLV) (S1-S2-S3-S4), calculated with the “Hepatic VCAR” segmentation software on the Whole-body CT scan.

**Figure 4 jcm-11-05287-f004:**
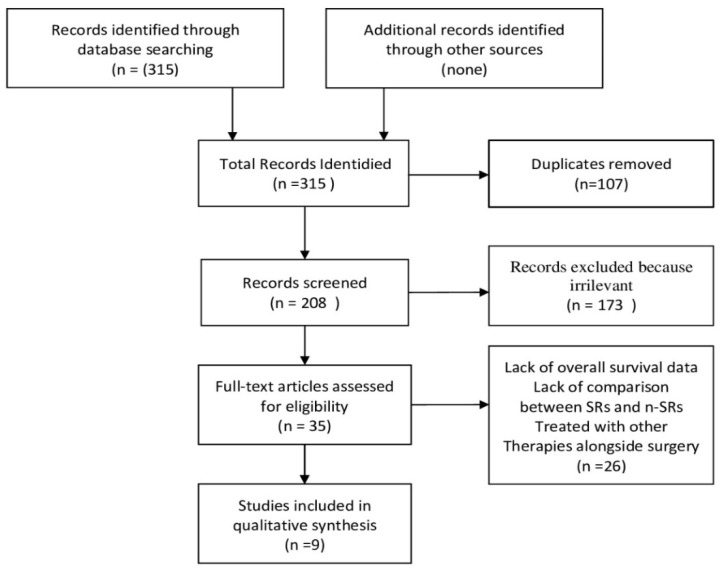
PRISMA flowchart.

**Table 1 jcm-11-05287-t001:** Characteristics of the studies.

Reference	Year	Number of Patients		Mean Age (Years)		Male		Median Survival (Months)		1-2-3 Years Survival Rate (%)		Study Type
		SR	n-SR	SR	n-SR	SR	N-SR	SR	n-SR	SR	n-SR	
Hamaoka [15]	2017	7	43	66	64	7	40	38	13.1	100; 100; 71	50; 20; 18	Retrospective
Ryon [16]	2020	21	186	55.2	61.2	18	151	19	5.8	N/A	N/A	Retrospective
Peng [17]	2011	201	402	55	55	187	374	20	13	42; 14.1; 11.1	37.8; 7.3; 0.5	Retrospective
Zheng [18]	2016	96	134	51.9	51.6	75	98	N/A	N/A	86.5; 0.4; 33.3	77.6; 47.8; 20.9	Retrospective
Kokudo [19]	2016	2093	4381	63.2	66.4	1744	3490	34.4	13.3	74.8; 49.1; 39.1	53.1; 25.3; 16.0	Retrospective
Tang [20]	2013	186	185	48.4	49.7	166	155	10	12.3	51.6; 28.4; 19.9	40.1; 17.0; 13.6	Retrospective
Fan [21]	2005	24	53	55	58	20	49	10	7.3	22.7; 9.8; 0	11.8; 0; 0	Prospective
Ye [22]	2014	90	75	49	45	81	80	8.2	7	28; 20; 15	17.5; 0; 0	Prospective
Zhang [23]	2015	28	56	47	51	27	54	15.6	9.1	66.5; 37.4; 28.5	32.3; 18.7; 15.6	Prospective

N/A: Not available.

**Table 2 jcm-11-05287-t002:** Characteristics of the patients.

Reference	Treatment	Child-Pugh		Portal Hypertension Yes (%)	AFP	PVTT Type Ⅰ(%)/Ⅱ(%)/Ⅲ(%)/Ⅳ(%)	HBV (%)
		A	B				
Hamaoka [15]	SR	7	0	N/A	0.321	N/A; N/A; 3(42.8); 4 (57.1)	42.8
	TACE, RFA, HAIC	32	11	N/A		N/A; N/A; 18(41.8); 25(58.1)	32.5
*p* Value		0.130		N/A		0.830	0.728
Ryon [16]	SR	12	6	N/A	0.027	N/A	81
	TACE, RFA, Y90 radioembolization, systemic therapy	67	85	N/A		N/A	83.1
*p* Value		0.171		N/A		N/A	0.750
Peng [17]	SR	197	4	N/A	0.07	27(14.4); 69 (34.3); 83 (41.2); 23(11.4)	85.5
	TACE	389	13	N/A		54(13.4); 136(33.8); 166(41.2); 46(11.4)	88.5
*p* Value		0.447		N/A		1.00	0.755
Zheng [18]	SR	75	21	85.4	0.816	25(26.0); 23(23.9); 23(23.9; 25(26.0)	58.3
	TACE	101	33	88		31(23.1); 32(23.8); 33(24.6); 38(28.3)	5.4
*p* Value		0.628		0.55		0.589	0.562
Kokudo [19]	SR	1877	216	12.8	<0.001	893 (42.7); 528 (25.2); 466 (22.3); 206 (9.8)	29.0
	TACE, RFA, HAIC, Systemic therapy	2512	1869	41.0		879 (20.1); 947 (21.6); 1476 (33.7); 1079 (24.6)	20.6
*p* Value		0.001		N/A		0.001	0.001
Tang [20]	SR	171	15	N/A	0.031	80 (43.0); 66 (35.5); 40 (21.5); N/A	85.4
	RFA+TACE	169	16	N/A		72 (38.9); 64 (34.6); 49 (26.5); N/A	80.5
*p* Value		0.49		N/A		0.687	0.513
Fan [21]	SR	18	6	N/A	N/A	N/A	N/A
	Systemic therapy	39	14	N/A		N/A	
*p* Value		0.351		N/A		N/A	N/A
Ye [22]	SR	84	6	4.6	0.704	N/A	13.3
	TACE	78	9	44.1		N/A	20.9
*p* Value		0.601		0.161		N/A	0.702
Zhang [23]	SR	28	0	N/A	0.417	N/A	92.8
	TACE	53	3	N/A		N/A	80.3
*p* Value		0.457		N/A		N/A	0.203

N/A: Not available.

**Table 3 jcm-11-05287-t003:** Risk of bias assessment.

Reference	Baseline Confounding	Selection of Partecipants	Classification of Intervention	Deviation from Intended Intervention	Missing Data	Measurement of Outcomes	Selection of Reported Results	Overall Risk of Bias
Hamaoka [15]	Low	Moderate	Low	Moderate	Low	Low	Low	Moderate
Ryon [16]	Moderate	Moderate	Low	Moderate	Moderate	Low	Low	Moderate
Peng [17]	Low	Low	Low	Low	Low	Low	Low	Low
Zheng [18]	Low	Moderate	Low	Low	Moderate	Low	Low	Moderate
Kokudo [19]	Moderate	Serious	Moderate	Moderate	Serious	Low	Low	Serious
Tang [20]	Moderate	Moderate	Low	Moderate	Moderate	Low	Low	Moderate
Fan [21]	Serious	Moderate	Moderate	Serious	Moderate	Low	Moderate	Serious
Ye [22]	Moderate	Moderate	Moderate	Seious	Moderate	Moderate	Low	Serious
Zhang [23]	Moderate	Moderate	Low	Moderate	Low	Moderate	Moderate	Moderate

## Data Availability

Not applicable.

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
