# Peer review of "Surgical Resection of a Recurrent Hepatocellular Carcinoma with Portal Vein Thrombosis: Is It a Good Treatment Option? A Case Report and Systematic Review of the Literature"

_jcm, 2022, doi:10.3390/jcm11185287_

Round 1
Reviewer 1 Report
This is an intersting case presentation and review on a contrversial topic. I have some comments regarding methodology and results:
1) I would add the case presentation to the results.
2) most of the studies were retrspective. despite thast the authors claim that the studies were of high quality according to the NOS and GRADE.
At least in NOS comparability is an important dimension of the exploration and for this work also pivotal. As shown in Table 2 patients had different CTP Scores I would suggest the authors explain more why they used the NOS and GRADE and not a simple Bias Analysis.
3) Regarding Table 2 please show the differences (p-value) CTP, portal hypertension, PVTT type and HBV status where possible.
4) In my opinion a meta-analysis could have been performed but with a strong reference to the high heterogeneity
3)
Reviewer 2 Report
The paper is very well written and very interesting.
However, there are some minor corrections that should be made:
Sorafenib is not considered a curative strategy for HCC and now is not the only first-line (lenvatinib is also).
The follow-up of the patient is very low so a bigger follow-up is needed.
There are several papers evaluating the possibility of surgical resection in PVT. Although, papers are still not conclusive and the survival rates vary a lot. Therefore, the conclusion should be more careful as this strategy is only appropriate in very selective patients and in places with an experient surgical team.
Reviewer 3 Report
The Authors report the case of a cirrhotic patient with recurrent HCC with tumor thrombus who was treated with a right hepatectomy followed by 4 months recurrence-free. In parallel, they provide a systematic review of the literature on the topic.
In general, the quality of the writing is acceptable, with some informal sentences and some redundancy that can be easily trimmed. I provided only a few examples on the pdf but the same principles should be applied to the whole manuscript.
Regarding the case report, I have several reservations:
- I have made some comments directly on the pdf that I'm attaching here
- The case is not thoroughly described and several aspects would need to be expanded and/or clarified
- the follow-up is short for concluding that the treatment was a good option, as reflected in the title. The Authors referred to the previous European guidelines, thus comparing the survival of their patient with that of sorafenib only. The recent guidelines have made substantial changes to the recommendations for systemic treatment agents with improved survival compared to the past. Therefore, I would suggest, the case report could be separated from the systematic review and presented separately after further follow-up.
Regarding the systematic review, I have the following observations:
- Review question and inclusion criteria adopted reflect PICO framework, although the elements of the "PICO" were not described in detail in a specific section of the report.
- The Authors have not stated that they have previously published a protocol for conducting the review (e.g. PROSPERO).
- The Authors justified in the text the exclusion from the review of potentially relevant studies, as they did not provide a complete list of all studies excluded with their relative motivation
- A summary table of risk of bias assessments was not edited for all included studies. Furthermore, the Authors did not use a satisfactory technique to assess the risk of bias in the individual studies included in the review. ROBINS-I tool (“Risk Of Bias In Non-randomized Studies - of Interventions”) is a more appropriate critical appraisal guide for evaluating the risk of bias in the results of NRSIs (Non-randomized Studies of the effects of interventions) that compare the health effects of two or more interventions.
- The impact of the risk of bias and the potential sources of heterogeneity of the studies were not discussed in the results interpretation.
- As a result, studies included are greatly heterogeneous, leading to the comparison of substantially different types of patients and different outcomes, which is a relevant bias for a systematic review.

Round 2
Reviewer 1 Report
The authors have adressed all my concerns.